# LEARNING TO STYLIZE SOUNDSCAPES FROM IN-THE-WILD VIDEOS

## ABSTRACT

Speech recordings convey a great deal of information about the scenes, resulting in a variety of effects ranging from reverberation to additional ambient sounds. In this paper, we learn to restyle input speech to sound as though it was recorded within a different scene, given an audio (or audio-visual) example recorded from that scene. Our model learns through self-supervision, taking advantage of the fact that natural video contains recurring sound events and textures. We extract an audio clip from a video and apply speech enhancement. We then train a latent diffusion model to recover the original sound, using another audio-visual clip taken from elsewhere in the video as a conditional hint. Through this process, the model learns to transfer the conditional example's sound properties to the input sound. We show that our model can be successfully trained using unlabeled, in-the-wild videos, and that an additional visual signal can improve its sound prediction abilities.

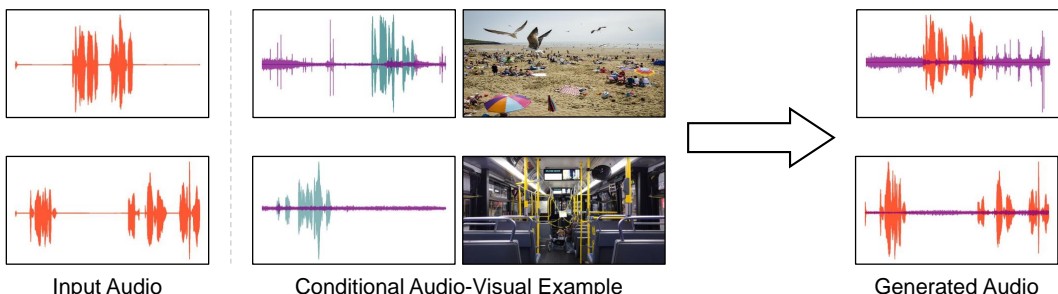

Figure 1: **Audio-Visual Soundscape Stylization**. We manipulate the soundscapes of input audio, guided by a user-provided audio-visual sample that specifies the desired auditory atmosphere, such as the sounds of people wandering about the beach or seated on a bus. This could involve elements like roaring ocean waves or echoing conversations. The ultimate goal is to generate **ambient soundscapes** that mimic those in the conditional examples, while allowing for some modifications to the **acoustic soundscapes** of the input audio. **We encourage the reader to watch and listen to the results on our project webpage.**

## 1 INTRODUCTION

Speech conveys a great deal about the scene that it was produced in, from the material properties of the scene's surfaces to its ambient sounds. A major goal in sound generation has been to accurately resynthesize speech to sound as though it were recorded in a given input scene, capturing these subtle scene properties.

Yet existing methods are often limited to reproducing only certain aspects of a scene's soundscape. These methods are often centered around capturing *acoustic* properties, like room impulse responses and spatial sound (Chen et al., 2022; 2023a; Richard et al., 2020), neglecting the holistic reproduction of ambient sounds that contribute to the unique auditory identity of a place. Take, for instance, a stroll along the beach (Figure 1): one may experience the whispers of the wind, the cries of seagulls, the laughter of children, and the lapping of waves – all harmonizing to shape the distinctive sound texture (McDermott & Simoncelli, 2011) of this environment. Furthermore, while there are alternative methods that are able to synthesize ambient sound effects based on textual instructions (Kreuk et al., 2023; Liu et al., 2023; Huang et al., 2023b), they often fail to take into account acous-

tic properties, making them less adaptable to real-life scenarios. Finally, existing methods largely require simulated or labeled training data, and cannot learn from abundantly available "in the wild" videos.

We take inspiration from classic work on sound textures (McDermott & Simoncelli, 2011), which generates sounds by matching their statistical properties to those of a conditional example. We pose the problem as *audio-visual soundscape stylization*: given a conditional audio (or audio-visual) example, we seek to transfer its ambient sound and acoustic properties to an input sound. We propose a simple, self-supervised learning task that exploits the fact that videos contain recurring sound textures (Du et al., 2023). This redundancy can be used as a free learning signal for stylization. We randomly sample two nearby audio-visual clips from a video, and remove soundscape-specific attributes from one of them performing speech enhancement. We then train a model to reverse this speech enhancement process, using the other audio-visual clip as a conditional "hint". In order to perform this task, the model needs to estimate the acoustic and ambient properties of the soundscape, and to successfully transfer them to an input example. At test time, we can then provide the model with a conditional example from a *different* video, and transfer its properties.

To implement our approach, we extend latent diffusion models (Rombach et al., 2022) to stylize input audio based on the provided audio-visual conditional example. We then train the model on a dataset of egocentric videos collected from the internet. After training, we assess our model's performance by synthesizing ambient sounds and reverberations in response to audio-visual cues. Through quantitative evaluations and perceptual studies, we show that:

- Unlabeled egocentric videos provide supervision for learning and transferring sound textures.
- Cross-modal information from sight improves sound stylization.
- Our model learns to stylize sound textures from in-the-wild audio-visual data, including ambient and acoustic soundscapes.

## 2 RELATED WORK

**Sound generation from visual and textual inputs.** Generating sound from visual and textual inputs has recently attracted much research attention. For visual-based methods, researchers have explored generation of sound effects, music, ambient sound, and speech from visual cues such as object impacts (Owens et al., 2016), musical instrument playing (Koepke et al., 2020), dancing (Gan et al., 2020; Su et al., 2021), and lip movements (Ephrat & Peleg, 2017; Prajwal et al., 2020; Hu et al., 2021). Different from these methods which focus on generating a specific type of sound from visual input, our method is centered on the stylization of soundscapes to seamlessly harmonize with their environmental context, guided by conditional audio-visual examples.

For text-based methods, Yang et al. (2023a) introduced a discrete diffusion model for generating ambient sound from text descriptions. Kreuk et al. (2023) employed VQGAN (Esser et al., 2021) for sound generation. Recently, latent diffusion models (Liu et al., 2023; Huang et al., 2023b) have made significant strides and delivered improved generations. All these methods necessitate text annotations to establish text-audio pairs, whether at the training (Kreuk et al., 2023) or representation (Liu et al., 2023; Huang et al., 2023b) level, which can be labor-intensive and limited in expressiveness with plain text descriptions. Our method uniquely stylizes soundscapes conditioned on audio-visual examples, obviating the need for annotations. Users can easily select their desired soundscape by looking and listening, thereby providing a complementary learning signal to text-based methods.

**Stylization in image and audio.** The concept of stylization was introduced in Hertzmann et al. (2023) which aimed to restyle input images based on a single user-provided image and its stylization. Various content types have been explored for image stylization, including image (Huang & Belongie, 2017; Isola et al., 2017; Zhu et al., 2017), text (Dong et al., 2017; Bau et al., 2021; Brooks et al., 2023), sound (Li et al., 2022b; Lee et al., 2022), and touch (Yang et al., 2022; 2023b).

Lately, researchers have delved into audio stylization tasks such as voice conversion (via feature disentanglement (Ulyanov, 2016; Verma & Smith, 2018) or adversarial learning (Kaneko & Kameoka, 2018; Li et al., 2021)), music timbre transfer (Huang et al., 2018), text-based audio editing (Wang et al., 2023), visual-based acoustic matching (Chen et al., 2022; 2023a), and audio effects stylization (Steinmetz et al., 2022). In contrast to prior works which focus sorely on ambient soundscapes generation or acoustic soundscapes manipulation, our approach seamlessly handles both scenarios.

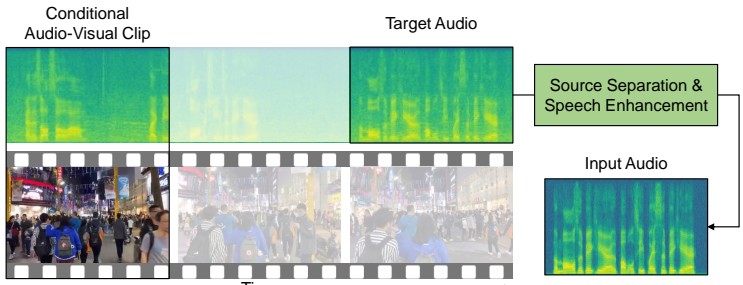

Figure 2: **Pretext task for conditional soundscape stylization**. We randomly select two distinct clips from a lengthy video, designating one as the conditional example and the soundtrack of the other as the target audio. Following this, we enhance the target audio, which is then employed as the input audio. Our model's objective is to stylize the input audio to closely emulate the target one, guided by an audio-visual clip, facilitating the semantic alignment between the input audio and the conditional example.

It offers an end-to-end solution for generating contextually appropriate soundscapes based on user-provided audio-visual examples. Moreover, unlike existing methods that rely on synthetic data due to the lack of paired stylization data, we collect a new dataset of real-world internet videos, providing a broader range of sound textures compared to synthetic alternatives.

**Audio-visual learning.** The natural correlation between audio and visual frames in videos has facilitated extensive audio-visual research, including representation learning (Arandjelovic & Zisserman, 2017; Korbar et al., 2018; Owens & Efros, 2018; Patrick et al., 2021; Morgado et al., 2021; Huang et al., 2023a), source separation (Zhao et al., 2018; 2019; Ephrat et al., 2016; Gao et al., 2018), audio source localization (Chen et al., 2021b; Harwath et al., 2018; Chen et al., 2023b), audio spatialization (Gao & Grauman, 2019; Morgado et al., 2018; Yang et al., 2020), visual speech recognition (Afouras et al., 2018), deepfake detection (Feng et al., 2023), and scene classification (Chen et al., 2020; Gemmeke et al., 2017). Inspired by this line of work, we introduce a novel task called *audio-visual soundscape stylization*, aiming to generate soundscapes harmonized with user-provided audio-visual examples.

## 3   AUDIO-VISUAL SOUNDSCAPE STYLIZATION

The objective of this task is to apply stylization to an input audio, harmonizing it with a conditional audio-visual example. We aim to train a mapping function denoted as $\mathcal{F}_\theta(\boldsymbol{a}_q, \boldsymbol{a}_c, \boldsymbol{i}_c)$, parameterized by $\theta$, which is responsible for generating soundsacpes for an input audio $\boldsymbol{a}_q$. This generation process is conditioned on a reference audio $\boldsymbol{a}_c$ and its corresponding image $\boldsymbol{i}_c$. In the following sections, we detail the training procedure for $\mathcal{F}_\theta$ using unlabeled data and our model for conditional soundscape stylization.

### 3.1   PRETEXT TASK FOR CONDITIONAL STYLIZATION

Our pretext task draws inspiration from the observation that sound textures within a video tend to exhibit temporal coherence (Du et al., 2023), especially when sound events are recurrent. Moreover, similar sound events often share semantically similar visual appearances (Owens et al., 2016; Huh et al., 2023), particularly when sourced from the same video. Consequently, randomly selected pairs of clips from a video frequently exhibit a notable correlation. In such instances, the model can leverage this correlation to enhance its stylization capabilities. Hence, we pose our task as an audio-to-audio stylization problem, where another clip from a different time step is provided as the conditional reference.

It is important to emphasize that this task goes beyond a trivial "copy and paste" operation. Our objective extends to the manipulation of not only ambient soundscapes but also acoustic counterparts, including factors like reverberation. Furthermore, our aim is to foster the model's capacity to seamlessly integrate these two soundscapes. To achieve this during training, we start by sampling two non-overlapping clips from a long-form video, centered at times $t$ and $t'$. One of these clips serves as

Figure 3: **One-to-many soundscape stylization at test time**. After training, our model is able to vary its output given different audio-visual examples. Here is an example where the input audio is stylized to emulate street and subway environments, including texture alterations in both ambient and acoustic soundscapes.

the conditional example, denoted as $\boldsymbol{a}_c$ and $\boldsymbol{i}_c$, while the soundtrack of the other clip is designated as the target audio $\boldsymbol{a}_o$ (Figure 2). Subsequently, we apply both a pre-trained source separation model, as depicted in (Petermann et al., 2022), to isolate the foreground speech, and a pre-trained speech enhancement model, as detailed in (Adobe, 2023), to further refine it. This process culminates in the generation of high-fidelity speech $\mathcal{H}(\boldsymbol{a}_q)$, which sounds as if it were recorded in a soundproofed studio. Please refer to A.3 for qualitative analysis of different enhancement methods. For preprocessing, we use an off-the-shelf voice activity detector (Silero, 2021) to ensure that each selected audio clip is more likely to contain speech.

Finally, the model is tasked with stylizing the input audio to closely resemble the original audio given the conditional audio-visual clip. This can be accomplished by minimizing a loss function $\mathcal{L}$, as expressed below:

$$\mathcal{L}_\theta = \|\boldsymbol{a}_o - \mathcal{F}_\theta(\boldsymbol{a}_e, \boldsymbol{a}_c, \boldsymbol{i}_c)\|_2^2 \,, \tag{1}$$

where $\boldsymbol{a}_e = \mathcal{H}(\boldsymbol{a}_q)$ is the input audio that has undergone source separation and speech enhancement.

We empirically find that the model tailors its stylizations according to the conditional examples, which aligns with the assumption that the conditional example is instructive for the input audio. At test time, we retain the flexibility to substitute the conditional example with a completely different audio-visual clip, enabling the potential for one-to-many stylization (Figure 3).

## 3.2 CONDITIONAL SOUNDSCAPE STYLIZATION MODEL ARCHITECTURE

We describe our conditional soundscape stylization model $\mathcal{F}_\theta$ (Figure 4), which is designed to stylize input audio based on a conditional audio-visual pair and consists of three main components: i) compressing the input audio into a latent space; ii) applying audio stylization using the conditional latent diffusion model; iii) reconstructing the waveform from the latent space.

**Adding noise to the input.** We propose to add Gaussian noise $\boldsymbol{n} \sim \mathcal{N}(\boldsymbol{0}, \boldsymbol{\sigma}^2)$ to the enhanced audio at both training and test time. This mixed audio is then employed as the input audio. The primary purpose of this is to mitigate the effect of "audio nostalgia" (Donahue et al., 2023). In other words, this noise helps conceal subtle artifacts of the original sound textures that may persist in the enhanced audio, which could otherwise lead to a trivial solution for this pretext task. Specifically, when the model is trained on the enhanced audio, the generated soundscapes might exhibit a suspicious resemblance to the originals. Conversely, when clean speech is used as input, the output may largely replicate the input. We consider this addition of noise as a type of data augmentation.

**Compressing mel-spectrograms.** We employ a variational auto-encoder (VAE) (Kingma & Welling, 2013) to compress the mel-spectrogram $\text{STFT}(\cdot) \in \mathbb{R}^{T \times F}$ into a latent space $\boldsymbol{z_0} \in \mathbb{R}^{T/r \times F/r \times d}$. Here, $\text{STFT}(\cdot)$ represents the short-time Fourier transform that converts the waveform into a mel-spectrogram, $r$ denotes the compression level, $T/r$ and $F/r$ is a lower-resolution time-frequency bin, and $d$ represents the embedding size at each bin. This compression is learned through training an auto-encoder to reconstruct sounds from a dataset, where the bottleneck serves as the encoded latent. Our auto-encoder comprises an encoder $\text{Enc}(\cdot)$ and a decoder $\text{Dec}(\cdot)$ using a stack of ResNet (He et al., 2016) and is trained by maximizing evidence lower-bound (ELBO) and minimizing adversarial loss. For our experiments, we adopt a pre-trained audio VAE model from Liu et al. (2023).

**Conditional audio-visual representations.** The conditional audio-visual example is represented using its latent vector. We employ separate audio and image encoders, denoted as $\mathcal{E}_a(\cdot)$ and $\mathcal{E}_i(\cdot)$, to extract audio embeddings $\mathcal{E}_a(\boldsymbol{a}_c) \in \mathbb{R}^L$ and image embeddings $\mathcal{E}_i(\boldsymbol{i}_c) \in \mathbb{R}^L$, where $L$ represents the embedding size. We initially use a ResNet-18 backbone (He et al., 2016) for audio and image representation respectively, but also explore alternative pre-trained encoders like CLIP (Radford et al., 2021) and CLAP (Elizalde et al., 2023). Prior to fusion, we apply linear projections to the image and audio embeddings, followed by feeding them into the diffusion model through cross-attention mechanism (Vaswani et al., 2017).

**Conditional latent diffusion model.** We train a conditional latent diffusion model to stylize soundscapes based on conditional audio-visual examples. Building upon the denoising diffusion probabilistic model (Ho et al., 2020) and the latent diffusion model (Rombach et al., 2022), our model breaks the generation process into $N$ conditional denoising steps, and improves the efficiency and quality of diffusion models by operating in the latent space.

Unlike generative diffusion models, our stylization model is seen as a conditional latent diffusion model that takes the encoded latent of the input audio and the conditional audio-visual example as conditions. More specifically, given the encoded latent of the original audio $\boldsymbol{z}_0 = \mathrm{Enc}(\boldsymbol{a}_o)$, the conditional audio-visual example $(\boldsymbol{a}_c, \boldsymbol{i}_c)$, and random noise $\boldsymbol{\epsilon} \sim \mathcal{N}(\boldsymbol{0}, \mathbf{I})$, our model first generates a noisy version $\boldsymbol{z}_t$ via a noise schedule (Song et al., 2020). We then define the training loss $\mathcal{L}_\theta$ by predicting the noise $\boldsymbol{\epsilon}$ added to the noisy latent, guided by the input audio $\boldsymbol{a}_e$ and the conditional audio-visual pair $(\boldsymbol{a}_c, \boldsymbol{i}_c)$. Thus we can rewrite $\mathcal{L}_\theta$ in Equation 1 as:

$$\mathcal{L}_\theta = \|\boldsymbol{\epsilon} - \boldsymbol{\epsilon}_\theta(\boldsymbol{z}_t, t, \boldsymbol{z}_e, \boldsymbol{a}_c, \boldsymbol{i}_c)\|_2^2 , \tag{2}$$

where $\boldsymbol{z}_e = \mathrm{Enc}(\boldsymbol{a}_e)$ is the encoded latent of the input audio $\boldsymbol{a}_e$.

**Classifier-free guidance.** Classifier-free guidance is a method commonly used in generative models, particularly diffusion models, to balance the trade-off between the quality and diversity of generated samples. This method involves jointly training the model for both conditional and unconditional denoising. During training, conditional examples are randomly nullified with a fixed probability (e.g., 10%). At test time, a guidance scale ($\lambda \geq 1$) is utilized to adjust the score estimates, aligning them with the conditional distribution and away from the unconditional distribution.

$$\tilde{\boldsymbol{\epsilon}}_\theta(\boldsymbol{z}_t, t, \boldsymbol{z}_e, \boldsymbol{a}_c, \boldsymbol{i}_c) = \lambda \cdot \boldsymbol{\epsilon}_\theta(\boldsymbol{z}_t, t, \boldsymbol{z}_e, \boldsymbol{a}_c, \boldsymbol{i}_c) + (1 - \lambda) \cdot \boldsymbol{\epsilon}_\theta(\boldsymbol{z}_t, t, \boldsymbol{z}_e, \phi, \phi) \tag{3}$$

We find this process enhances the output quality and relevance of stylized samples.

**Recovering the waveform.** Following the estimation of noise $\tilde{\epsilon}_\theta$ from the diffusion model, we retrieve the encoded latent of the stylized mel-spectrogram. This latent is then fed into the VAE decoder to reconstruct the stylized mel-spectrogram. Finally, a pre-trained HiFi-GAN vocoder (Kong et al., 2020a) is employed to reconstruct the waveform, as outlined in Liu et al. (2023).

## 4 EXPERIMENTS

### 4.1 EXPERIMENTAL SETUP

**Dataset.** We collected a new *CityWalk* dataset which includes egocentric videos with diverse ambient and acoustic soundscapes, recorded in places like trains, buses, streets, beaches, and shopping malls. We sourced these videos using the keywords "city walk+POV" from YouTube. Here, we selected a subset of 223 videos for training, with duration ranging from 5 to 225 minutes, totaling 150 hours. We ensure that these videos only contain naturally occurring sounds in the scenes, without any post-edited voice-overs or music. See Appendix A.1 for more dataset details.

**Model Configurations.** We use the VAE and HiFi-GAN vocoder from (Liu et al., 2023), which are trained on the combination of AudioSet (Gemmeke et al., 2017), AudioCaps (Kim et al., 2019), Freesound (Fonseca et al., 2021), and BBC Sound Effect (BBC, 2017) datasets. The VAE is configured with a compression level $r$ of 4 and latent channels $d$ of 8. For extracting audio and image embedding, we have two options: i) use a from-scratch ResNet-18 encoder; ii) utilize a fine-tuned CLAP audio encoder (Elizalde et al., 2023) derived from our audio-only model, alongside a fixed CLIP image encoder (Radford et al., 2021). These encoders are integrated into the diffusion model

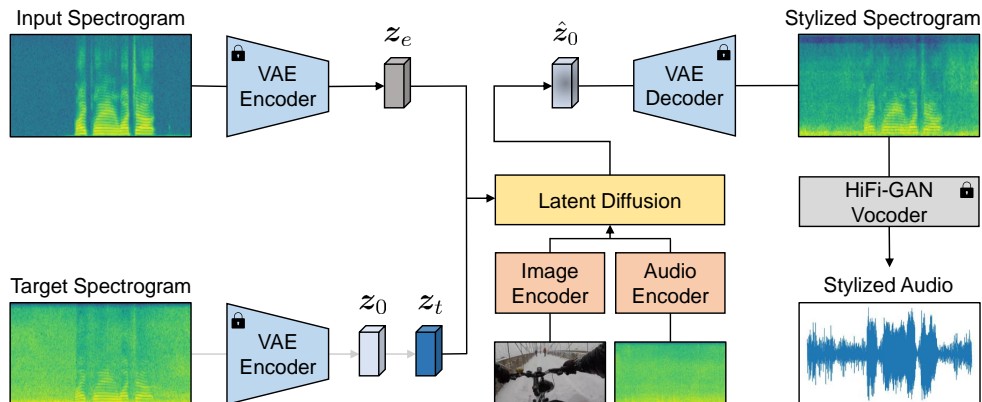

Figure 4: **Model architecture**. We stylize the input audio, conditioned on an audio-visual pair sampled from the same video. We encode both the input and target spectrograms to the latent space using a pre-trained VAE encoder, and feed them into a latent diffusion model together with the conditional audio-visual embedding, aiming to harmonize the encoded latent of the input spectrogram with the target one. Finally, the waveform is reconstructed from the latent space, employing a pre-trained VAE decoder followed by a pre-trained HiFi-GAN vocoder. Note that the VAE encoder for the target spectrogram is not used at test time.

through late fusion (Wang et al., 2020) and cross-attention (Vaswani et al., 2017). The diffusion model is based on a U-Net backbone, consisting of four encoder and decoder blocks with downsampling and upsampling and a bottleneck in between. Multi-head attention with 64 head features and 8 heads per layer is applied in the last three encoder and first three decoder blocks. During the forward process, we employ $N = 1000$ steps and a linear noise schedule, ranging from $\beta_1 = 0.0015$ to $\beta_N = 0.0195$, to generate noise. Additionally, we leverage the DDIM sampling method (Song et al., 2020) with 200 sampling steps. For classifier-free guidance, we set the guidance scale $\lambda$ to 4.5, as described in Equation 3.

**Training procedures.** To enhance training efficiency, we divide all videos into 10-second video and audio clips. We apply a frame-level voice activity detector (Silero, 2021) to the resulting audio clips to detect speech onset. Subsequently, we randomly select two 2.56-second audio clips from the same source – one for the target audio and the other for the conditional audio. The conditional image is chosen by randomly sampling one video frame within the scope of the selected conditional audio. Our model is trained using the AdamW optimizer (Loshchilov & Hutter, 2017) with a learning rate of $10^{-4}$, $\beta_1 = 0.95$, $\beta_2 = 0.999$, $\epsilon = 10^{-6}$, and a weight decay of $10^{-3}$ over 200 epochs.

**Evaluation metrics.** To assess the performance of our models, we use both objective and subjective metrics. Our objective metrics include the *Fréchet Audio Distance* (FAD) (Kilgour et al., 2019), *Fréchet Distance* (FD) (Liu et al., 2023), *Kullback-Leibler divergence* (KL), *Inception Score* (IS) (Salimans et al., 2016), and *Audio-Visual Correspondence* (AVC) (Arandjelovic & Zisserman, 2017). FD and FAD measure the similarity between real and generated audio using different classifiers (FAD employs VGGish (Hershey et al., 2017) and FD uses PANNs (Kong et al., 2020b)). KL quantifies the distributional similarity between real and generated audio, while IS evaluates the quality and diversity of generated audio. AVC assesses the correlation between audio and image, utilizing features extracted by either OpenL3 (Cramer et al., 2019) or ImageBind (IB) (Girdhar et al., 2023). In addition, we conduct a subjective evaluation through Amazon Mechanical Turk. Human participants are asked to rate audio generated by various methods based on its similarity to the soundscapes in the given audio-visual example. This rate considers four criteria: *overall quality* (OVL), *relation to ambient soundscapes* (RAM), *relation to acoustic soundscapes* (RAC), and *relation to visuals* (RVI), with scores ranging from 1 (low correlation) to 5 (high correlation). See Appendix A.2 for more human evaluation details.

**Baselines.** Given the challenge of accurately modeling acoustic soundscapes in outdoor scenes using the previous methods (Chen et al., 2022; Steinmetz et al., 2022), especially in our in-the-wild setup, we restrict our comparison to baselines that specialize in generating ambient soundscapes:

| Method | FD (↓) | FAD (↓) | KL (↓) | IS (↑) | AVC (↑) | | OVL (↑) | RAM (↑) | RAC (↑) | RVI (↑) |
| | | | | | IB | L3 | | | | |
|---|---|---|---|---|---|---|---|---|---|---|
| Ground Truth | / | / | / | / | 0.221 | 0.979 | 4.03 ± 0.09 | / | / | 4.15 ± 0.11 |
| Captioning (aud) | 14.30 | 9.73 | 1.09 | 1.49 | 0.082 | 0.799 | 2.58 ± 0.14 | 2.53 ± 0.12 | 3.08 ± 0.10 | 3.13 ± 0.07 |
| Captioning (sfx) | 12.53 | 9.12 | 1.14 | 1.51 | 0.091 | 0.806 | 2.77 ± 0.08 | 3.01 ± 0.07 | 3.18 ± 0.13 | 3.22 ± 0.12 |
| Captioning (img) | 17.27 | 9.24 | 1.30 | 1.46 | 0.079 | 0.788 | 2.14 ± 0.10 | 2.22 ± 0.14 | 3.07 ± 0.15 | 3.09 ± 0.10 |
| AudioLDM | 9.33 | 3.97 | 0.91 | 1.53 | 0.107 | 0.823 | 3.08 ± 0.13 | 3.03 ± 0.10 | 3.15 ± 0.11 | 3.12 ± 0.13 |
| Separate & Remix | 8.87 | 3.48 | 0.70 | 1.51 | 0.114 | 0.822 | 3.15 ± 0.11 | 3.29 ± 0.09 | 3.11 ± 0.06 | 3.38 ± 0.09 |
| Ours | **5.13** | **1.64** | **0.59** | **2.03** | **0.172** | **0.915** | **3.68 ± 0.14** | **3.72 ± 0.08** | **3.55 ± 0.09** | **3.59 ± 0.06** |

Table 1: Evaluation results on the *CityWalk* dataset. Captioning can be driven by original conditional audio (aud), separated sound effects (sfx), and conditional images (img). The subjective OVL, RAM, and RAC metrics are presented with 95% confidence intervals.

- **Captioning**: This cascaded approach employs pre-trained image (Li et al., 2022a) or audio (Mei et al., 2023) captioning models to generate captions from conditional examples, which are then used to generate sound effects using a text-to-audio model (Liu et al., 2023).
- **AudioLDM** (Liu et al., 2023): AudioLDM is originally used for text-to-audio synthesis. Here we switch the text condition with the audio one, allowing us to perform audio-to-audio analogies. We train a model using isolated sound effects instead of the original audio to enforce that the resulting audio does not include any speech, as we aim to avoid having two speakers in the final output.
- **Separate & Remix** (Petermann et al., 2022): This method adopts a simple "copy and paste" strategy. It uses a pre-trained source separation to isolate sound effects from the conditional audio and overlays them onto the input audio.

### 4.2 COMPARISON TO BASELINES

**Quantitative results.** We start by presenting the quantitative results of our model and baselines in Table 1. Our model, whether operating in an uni-modal (Table 3) or audio-visual (Table 1) condition, consistently outperforms three baselines across multiple objective metrics, including FD, FAD, KL, and IS. These results suggest that our model excels in generating more realistic soundscapes compared to the baselines. In particular, Separate & Remix is worse than our method, despite the fact that it receives nearly the same ambient soundscapes from the conditional audio. This is probably because our method can not only manipulate ambient soundscapes but also acoustic ones. We also find that all three Captioning methods perform worse than Separate & Remix, perhaps due to errors introduced by automatic captioning. Among the Caption-based methods, we observe that using separated sound effects produces more precise captions than the others, leading to the best performance. Moreover, although AudioLDM is trained to resemble the similar ambient soundscapes of the conditional audio, it still cannot achieve comparable performance to Separate & Remix, whereas our method can. This highlights the importance of acoustic soundscapes when it comes to soundscape stylization.

To further validate our model's performance, we conducted a human evaluation. We randomly selected 30 generated audio samples from the test set, with each sample scored by 20 participants. To prevent random submissions, we included one control set consisting entirely of noise. The results of the human evaluation consistently favored our model's stylized audio, as indicated in the last four columns of Table 1, which aligns with the objective evaluation results. Interestingly, we observed that the RAC metrics of all the baseline methods were on par with each other. This consistency could be attributed to the fact that their output is mixed with the same speech as the input, without considering the difference in acoustic soundscapes. This finding also emphasizes the importance of considering acoustic nuances in soundscape stylization, which our model effectively addresses.

**Qualitative results.** We visualize how our results vary under different conditional examples and compare our model with baselines in Figure 5. We also provide additional results in Appendix A.3. For the caption-based methods, we only present the best model, which relies on isolated sound effects. Notably, we observe that while these methods occasionally align with the provided conditional examples, they often falter in most instances (like the artifacts that occurred in the second output). This may be because the success of stylization strongly hinges on whether the captions are correct. The AudioLDM method is able to generate plausible ambient soundscapes that match those in the

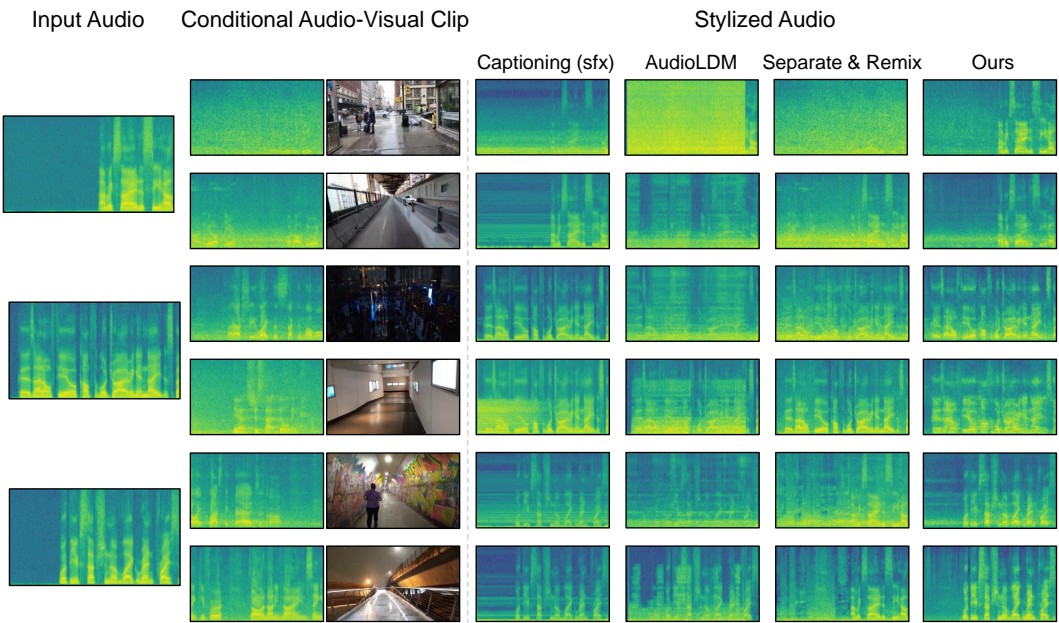

Figure 5: **Model comparison**. We show conditional soundscape stylization results between our method and baselines, where each input audio is conditioned on two different audio-visual examples.

conditional examples, but with apparent flaws when the conditional audio is more complex, such as the rainy street scenario depicted in the first example. For Separate & Remix, we find that it "copies and pastes" ambient sounds from the conditional example without considering the acoustic nuances of the soundscape. This is evident when, for instance, the conditional audio is situated in a tunnel or on a snowy street (the last two examples), resulting in echoed or hazy speech. This stark contrast underscores this method's limitations in terms of contextual awareness. Besides, the performance of the separation model tends to fall short, leading to speaker overlap in the output, as seen in the penultimate example. We encourage the readers to visit our project webpage to listen to our outputs and for more qualitative results. Please check out our results video in the supplementary materials and visit our project webpage at project webpage for additional qualitative results and to listen to our outputs.

## 4.3 ABLATION STUDY AND ANALYSIS

Table 2 presents the ablation studies on the *CityWalk* dataset. We analyze the following model variants: i) using the clean audio from the enhancement model as inputs instead of introducing Gaussian noise; ii) employing only the source separation model to isolate the target speech that preserves the original acoustic soundscapes (whereas our default method involves employing a speech enhancement model afterward to modify speech properties); iii) only using the conditional model (no CFG) to stylize input; iv) only using the unconditional model to stylize input; v) train a ResNet-based audio-visual encoder from scratch rather than using pre-trained CLIP and CLAP. Judging from the results, we draw the following observations:

**Adding noise mitigates audio nostalgia.** We ask whether adding noise will help mitigate the effect of "audio nostalgia" described in Section 3.2. Table 2 clearly demonstrates that our approach outperforms the model trained on clean speech by a large margin, providing empirical evidence to support our hypothesis

**Acoustic soundscapes play an important role in soundscape stylization.** We ask whether our method can resemble plausible acoustic soundscapes to the conditional examples. To examine this, we train a model using only the speech extracted from a separation model, which excludes the acoustic factor during stylization. As depicted in Table 2, the performance of this variant drastically declines, indicating the critical role of considering acoustic soundscapes in soundscape stylization.

| Method | FD (↓) | FAD (↓) | KL (↓) | IS (↑) |
|---|---|---|---|---|
| (i) Clean Input | 6.12 | 2.66 | 0.75 | 1.49 |
| (ii) Separation-only | 6.11 | 2.44 | 0.73 | 1.46 |
| (iii) No CFG | 7.44 | 3.37 | 0.79 | 1.28 |
| (iv) No Condition | 16.77 | 7.59 | 1.27 | 1.45 |
| (v) From Scratch | 6.18 | 2.41 | 0.74 | 1.51 |
| Ours-Full | **5.13** | **1.64** | **0.59** | **2.03** |

| A | V | FT | FD (↓) | FAD (↓) | KL (↓) | IS (↑) | IB (↑) | L3 (↑) |
|---|---|---|---|---|---|---|---|---|
| ✗ | ✓ | ✗ | 6.40 | 2.28 | 0.91 | 1.41 | 0.125 | 0.882 |
| ✗ | ✓ | ✓ | 6.39 | 2.29 | 0.91 | 1.40 | 0.123 | 0.884 |
| ✓ | ✗ | ✗ | 10.22 | 6.12 | 0.88 | 1.53 | 0.111 | 0.817 |
| ✓ | ✗ | ✓ | 5.94 | 2.08 | 0.71 | 1.74 | 0.137 | 0.892 |
| ✓ | ✓ | ✗ | **5.13** | **1.64** | **0.59** | **2.03** | **0.172** | **0.915** |
| Ground Truth | / | / | / | / | 0.221 | 0.979 | | |

Table 2: Ablation studies on the *CityWalk* dataset. Table 3: Comparison of our uni-modal (A or V) and audio-visual models. A: Audio; V: Visual; FT: Fine-tune.

**CFG and condition enhance the stylization relevance.** We assess the impact of CFG and conditioning on enhancing output relevance. Table 2 demonstrates that our approach outperforms variants that lack CFG or conditioning, demonstrating their effectiveness in enhancing output relevance.

### 4.4 COMPARISON TO UNI-MODAL MODELS

We explore the performance of CLAP and CLIP encoders across various conditional settings. Table 3 presents a comparison between the fine-tuned CLAP encoder and its non-fine-tuned counterpart. Notably, fine-tuning significantly enhances performance, suggesting that the original CLAP model has limited generalization capabilities for our dataset. Moreover, fine-tuning the CLIP encoder has little effect on performance gain, indicating its inherent strong generalization abilities. Based on these findings, we adopt a late fusion approach (Wang et al., 2020), combining a fine-tuned CLAP audio encoder with a fixed CLIP image encoder for our audio-visual model. This configuration achieves the best performance among all the models, showcasing the effectiveness of integrating audio and visual information to create more comprehensive soundscapes.

### 4.5 LIMITATION AND ANALYSIS

While our model shows promising results across various scenarios, it is important to note that its performance is sometimes inconsistent. In particular, our model struggles with certain challenges related to vocal effort (Hunter et al., 2020) and audio-visual synchronization (Chen et al., 2021a), making it hard to capture nuances such as pitch variations due to speaker-listener distance or maintaining audio consistency with moving objects in the visual scene (Figure 6).

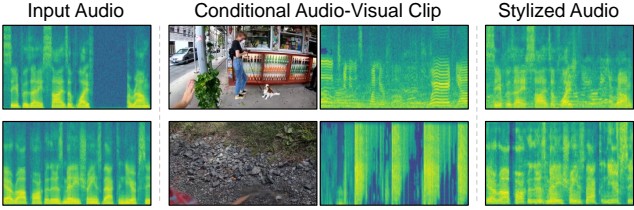

Figure 6: **Failure cases**. Our model fails to resemble the acoustic soundscapes of the conditional example, perhaps due to the vocal effort (up). It also fails to generate ambient soundscapes synchronized with the conditional example (bottom).

Additionally, our model may encounter difficulties when generating audio in unfamiliar contexts, highlighting the need for scaling up the model and dataset to enhance its generalization abilities. Nonetheless, considering the wealth of online videos available, this paper overall provides a solution for advancing the field of soundscape stylization within the audio-visual context.

## 5 CONCLUSION

In this paper, we present a novel task termed *audio-visual soundscape stylization*, aiming to faithfully replicate the complex soundscapes from unlabeled in-the-wild audio-visual data. Leveraging diffusion models in a self-supervised manner, our method successfully captures ambient and acoustic soundscapes that mimic user-provided audio-visual examples. Objective and subjective evaluations demonstrate our model's ability to capture the nuances of sound textures. We hope that our work not only contributes to audio-visual soundscape stylization but also encourages further exploration into how sound textures shape our perception of the world.

REPRODUCIBILITY STATEMENT

Our dataset curation details can be found in Section 4.1 of the main paper and Appendix A.1. The most salient training details and hyperparameters are also presented in Section 4.1 of the main paper. Since our proposed method is straightforward (Section 3.2 of the main paper), it ensures a high level of reproducibility for our work. Our code is also available in the supplementary material.

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

# A    APPENDIX

The Appendix is organized as follows: First, we provide additional details regarding dataset collection in Section A.1. Secondly, we describe how we conduct human study in Section A.2. Thirdly, we include additional qualitative examples about speech enhancement and model comparison in Section A.3.

## A.1    DATASET COLLECTION

We introduce the *CityWalk* dataset, a collection of egocentric videos for *audio-visual soundscape stylization*. This dataset features a rich diversity of real-world sound textures, encompassing both ambient soundscapes and acoustic soundscapes. The videos were gathered from YouTube, utilizing search queries such as "city walk+POV" and "city walk+binaural." The dataset comprises 3,447 untrimmed videos, with a total length of 2,395 hours. Detailed duration statistics are depicted in Figure 8. As illustrated in Figure 7, CityWalk contains a wide spectrum of audio recordings, including human speech and ambient sound, captured in varied environments such as urban streets, train stations, buses, beaches, airports, shopping malls, mountains, markets, boats, and churches, spanning diverse weather conditions.

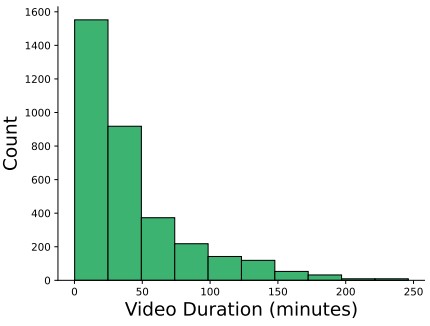

Figure 7: Duration distribution on the *City-Walk* dataset.

For data filtering and training in *audio-visual soundscape stylization*, we first split each video into 10-second clips and run a pre-trained YAMNet model (Plakal & Ellis, 2019) to tag each soundtrack. This step ensures the presence of the targeted audio types within these clips, confirming they have not been substituted with alternate sounds, such as voice-overs or background music. Furthermore, we use an off-the-shelf voice activity detector (Silero, 2021) to detect speech onsets and exclude silence intervals. The total duration of *CityWalk* is 1,150 hours. Due to computational constraints, we randomly sample 150 hours for training purposes, allocating 142 hours for training data and reserving the remainder for evaluation. Please note that the source of training videos and testing videos do not overlap.

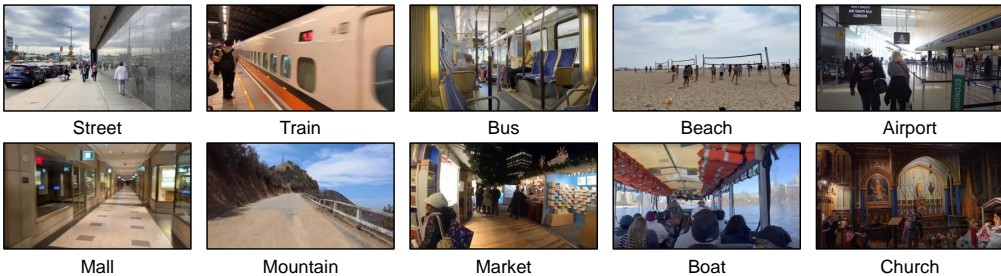

Figure 8: **Example frames of the *CityWalk* dataset**. We randomly select several scenes here for showcasing.

## A.2    HUMAN EVALUATION

For human subjective evaluations (i.e., OVL, RAM, and RAC) in Table 1, we developed an interface shown in Figure 9. We selected 30 test samples and each of them was rated by 20 participants who are native English speakers to ensure reliability. To maintain anonymity, we organized model outputs in a folder and assigned them with random identifiers. Participants were then tasked with rating each audio file within the context of an audio-visual example by completing the last four columns. We also included a control set containing only white noise to prevent random submissions. Our analysis of the control set revealed consistently low scores given by all human raters, reinforcing the reliability of our evaluation.

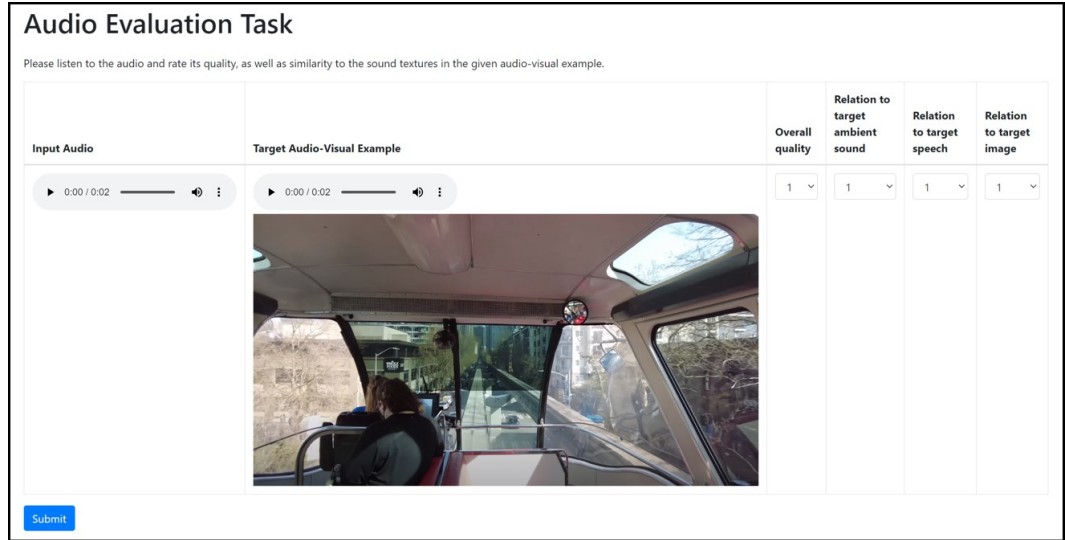

Figure 9: **Interface for Amazon Mechanical Turk**. We provide a screenshot of the interface designed for evaluating *audio-visual soundscape stylization*. Participants are instructed to complete the last four columns prior to advancing to the next example. Upon clicking the "Submit" button, participants will be navigated to the next question.

## A.3 ADDITIONAL RESULTS

**Additional speech enhancement visualization.** We propose a two-stage approach for speech enhancement, consisting of source separation (Petermann et al., 2022) as the initial step, followed by speech enhancement (Adobe, 2023) as the subsequent stage, to yield the final input audio. This approach is imperative because relying solely on either source separation or speech enhancement fails to yield the desired speech quality we require.

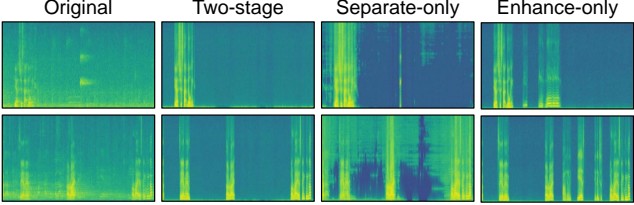

Figure 10: Qualitative comparison on different enhancement strategies.

As depicted in the penultimate column of Figure 10, using the separation model alone results in isolated speech that retains its original acoustic soundscapes, making our stylization model struggle to acquire the necessary acoustic traits during the training process. Besides, utilizing the enhancement-only method (as seen in the last column of Figure 10) often treats both close-talking and far-field speech as the enhanced target, leading to imprecise foreground speech identification. This, in turn, can precipitate a degradation in stylization quality.

Our proposed approach incorporates the strengths of both aforementioned strategies, employing a sequential workflow that initially separates and isolates foreground speech and subsequently refines it through enhancement (as shown in the second column in Figure 10). This ensures that the final input aligns with our requirements, balancing ambient and acoustic soundscapes.

**Additional qualitative comparisons.** In Figure 11, we present additional qualitative comparisons between our approach and the baselines. To provide a comprehensive evaluation, we employ the same held-out video clips at different time intervals as conditional examples, allowing us to illustrate our model's proficiency in reproducing the desired target audio. Furthermore, we introduce conditional examples devoid of speech to facilitate a more precise evaluation of our method and Separate & Remix.

Specifically, when confronted with non-speech conditional clips, Separate & Remix manages to extract the ambient soundscapes from the conditioning. However, it struggles to strike an appropriate

Input Audio    Conditional Audio-Visual Clip                    Stylized Audio                    Target Audio

Captioning (sfx)    AudioLDM    Separate & Remix    Ours

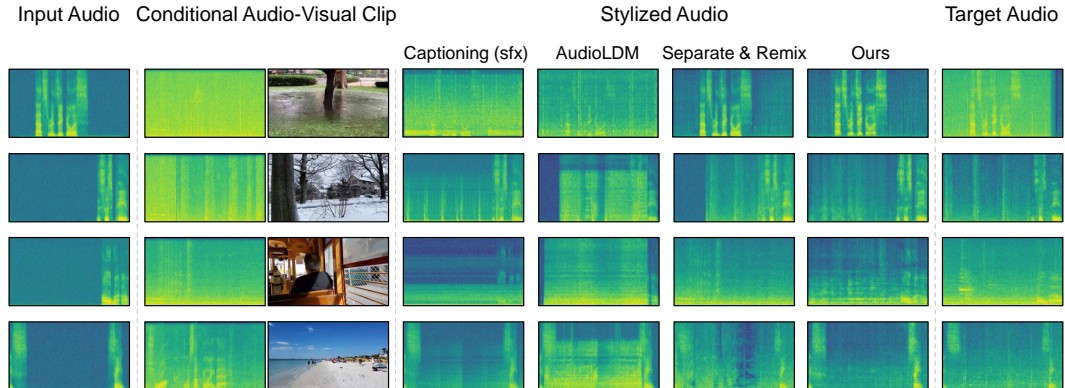

Figure 11: **Randomly selected qualitative results**. We present conditional examples derived from the same video as the input audio, but at different time steps. This is an extension of Figure 5 in the main paper.

balance in volume, resulting in the ambient sounds overwhelming the speech, as exemplified in the third case.

Captioning-based methods also face challenges when presented with conditional examples devoid of speech. Even in such instances, the generated captions often fall short of capturing the nuanced details within the input. For instance, in the second conditional example, where the conditional audio features footsteps on snow, the generated caption only identifies the presence of footsteps without acknowledging the snow. Consequently, the resulting sound effects deviate from the original conditional example.

Although AudioLDM can replicate ambient soundscapes to some extent, its quality is not as consistent as our approach, probably due to its heavy reliance on isolated ambient sound sources. Furthermore, we find that AudioLDM occasionally produces glitches in the audio, as evident in the last three examples.

We note that while our method generally outperforms these baselines by considering both ambient and acoustic soundscapes, there are cases where it appears to prioritize the input audio over the conditional one when generating ambient soundscapes. This may lead to the intensity of the generated soundscapes not matching that of the conditional examples.

For example, in the first conditional example of Figure 11, where the conditioning features heavy rain, our model stylizes the audio to resemble light rain instead, possibly influenced by the mild tone (whisper) of the input audio. This suggests that our model may place more emphasis on the acoustic soundscapes during the stylization process, resulting in such deviations.

