# OpenReview forum: "Learning to Stylize Soundscapes from In-the-Wild Videos"
_ICLR.cc/2024/Conference — ICLR 2024 Conference Withdrawn Submission_

### Official Review · Reviewer_V8de · 2023-10-29

**Soundness:** 2 fair
**Presentation:** 3 good
**Contribution:** 3 good
**Rating:** 5
**Confidence:** 3

**Summary:**

This paper presents a method to perform audio-visual soundscape stylization, which aims to authentically recreate intricate sound environments using unlabeled, real-world audio-visual data. Employing self-supervised learning and diffusion models, the method extracts audio from videos, applies speech enhancement, and trains a latent diffusion model to regenerate the original sound by utilizing other audio-visual clips from the same video as conditional inputs to the generative model. The model learns to transform the input sound to resemble recordings from different scenes, demonstrating its ability to transfer sound properties. The experiments of this paper shows that the proposed approach could be a valid way to perform audio-style transfer with audio-visual conditioning.

**Strengths:**

1. The paper is generally speaking well written, the ideas are presented in a succinct and nice way, the figures are well displayed and they also show very clearly how the proposed system operates (esp. Figs 3, 4).

2. It is really important that the authors also collected this new dataset called CityWalk that could be extremely useful for other researchers in the field.

**Weaknesses:**

- My main concern about this paper is that I do not see any fair comparison with the models in the literature that also perform audio-style-transfer in a similar fashion. Specifically, it is surprising that Separate & Remix (Petermann et al., 2022) would perform so comparably with the model proposed here which is trained specifically for this task and is much more computationally heavy. I believe that the remixing strategy of the aforementioned baseline could be fine-tuned for each example in a more proper way. For instance, it is evident that in Figure 5, first row, the Input-audio is remixed at a very low signal-to-noise ratio (SNR) for the Separate & Remix column which would probably make it sound worse. Or for example in the same Figure 5, final row, Separate & Remix seems to not be embedding the full input-audio, which clearly does not make sense, since the separation model should be simply run on the conditional audio input, extract the background noise and then remix the background noise with the input speech. Thus, I consider that by fixing those issues, a proper baseline model could be used to compare against this method.

- Building upon my previous argument, the proposed method is composed of multiple elements and many more millions of trainable parameters and extra computational complexity compared to other baselines with much simpler models as for example Separate & Remix (Petermann et al., 2022). Thus, the models should either try to match the computational and memory complexity and perform the same experiments for the baseline models or at least use the Separate & Remix (Petermann et al., 2022) as an input condition to the model to at least show that they can further enhance the performance of the overall model.

- The authors throughout the text try to make certain that their method is general enough to be applied to any type of sound, but it seems to me that the authors have only tested their model with speech style transfer, thus, I think they should be more explicit in what exactly the application of this paper is. Moreover, the pre-trained vocoders are not self-supervised, and thus, in a similar sense, the authors should try to avoid using this characterization or at least try to be more explicit on which parts were self-supervised during training.

I would be more than happy to increase my score if all the above weaknesses are addressed by the authors.

**Questions:**

Do you think that this method could be learned in a totally unsupervised way, meaning that there are no labeled data to train even vocoders?

---

### Official Review · Reviewer_UAzL · 2023-10-30

**Soundness:** 2 fair
**Presentation:** 2 fair
**Contribution:** 1 poor
**Rating:** 3
**Confidence:** 3

**Summary:**

The paper proposes a self-supervised approach for transferring vanilla speech sounds into styles of in-the-wild videos by considering both environment acoustics and ambient sounds. The model consists of a VAE for autoencoding sound spectrogram into latent space, and a conditional latent diffusion model for transferring latent feature of raw speech into latent feature of the speech under the target environment, where the condition takes the latent feature of both input image and target sound as input. The output of VAE is transferred to a spectrogram and then converted into waveform by HiFiGAN. Experiments show that the proposed method outperforms several  text-to-audio baselines.

**Strengths:**

1. The paper writing is clear and easy to follow.

2. The paper contributes a dataset for the task of in-the-wild soundscape transfer, which is potentially useful to the research community.

**Weaknesses:**

1. The model proposed by authors is quite straightforward without enough technical novelty, most of them are existing components like the VAE and latent diffusion models for style transfer, and the HiFiGAN for vocoder.

2. The baselines which the method compares against are not strong, these baselines are not targeted towards soundscape style transfer. I think at least Chen et al.2022 should be compared as the model architecture is easy to implement.

3. I feel like the quality of style transferred results shown in the demo website are not good enough, and some examples after transferring lost too much clarity in speech, and some examples don’t match the environment acoustic property.

4. The appendix provides some details of user study but is not enough. What is the distinct number of the voters, their qualification level like historical approval rate, how much time on average did they spend for each vote etc?

**Questions:**

See weakness.

---

### Official Review · Reviewer_HkqM · 2023-11-01

**Soundness:** 2 fair
**Presentation:** 2 fair
**Contribution:** 1 poor
**Rating:** 3
**Confidence:** 4

**Summary:**

The paper introduces an approach to perform soundscape style-transfer, aiming to replicate the soundscape of a in-the-wild audio-visual data. The method employs diffusion models in a self-supervised manner to capture ambient and acoustic soundscapes. The authors conduct experiments with baseline systems and the proposed approach trained with the newly proposed dataset, and also conducts multiple ablation studies.

**Strengths:**

- The authors have undertaken the challenging task of matching soundscapes from in-the-wild videos, incorporating both audio and visual information. This opens up possibilities for utilizing large-scale dataset with self-supervised learning, particularly in real-world environments.
- The authors created a new audio-visual video dataset which is valuable to this research field.

**Weaknesses:**

- The proposed method’s pipeline is very much utilizing methods from previous works, making it challenging to discern the authors’ highlighted contributions throughout the manuscript.
- It’s unclear why the model needs to take both the enhanced signal $a_e$ and the original target signal $a_0$ as the input. Since the soundscape information should be captured from $a_c$ not $a_0$ (where the soundscape of $a_c=a_0$), the objective of the proposed system is to just predict the noise applied on $z_t$. With this approach, the model doesn’t need to encode any information from $a_c$ or $i_c$, where it should focus on capturing the soundscape information from them.
- Regarding objective evaluation: With the current approach, the model can be evaluated on reconstruction task as Equation (1). Yet, the paper only uses evaluation metrics related to generation task. The reviewer suggests evaluating on intrusive objective metrics to focus more on acoustic matching rather than the generative approach.

**presentation feedbacks**

- Table 1 is misleading. All the captioning-based models also use AudioLDM, which looks conflicting with the AudioLDM on the Table.
    - The choice of AuidoLDM and Separate & Remix as baseline system is puzzling, as these systems are not suited for the task of generating or matching reverberation applied on input speech signal. To compare with these methods, the proper way would be to remove all speech signals and compare only with the soundscape audio, and to perform acoustic matching with different baselines.
- Table 3 is misleading since the final version of model uses the fine-tuned model of CLAP and not fine-tuned model of CLIP. The reviewer suggests adding a column to distinguish FT-A and FT-V in the table for better clarity.
    - In Section 3.2 **Conditional audio-visual representations**, the manuscript mentions using a ResNet-18 backbone for audio | image representation. Maybe it’s better to remove this sentence since the final model is using CLAP & CLIP?

**Questions:**

- Will the authors release the *CityWalk* dataset?
    - How did the authors handle the data imbalance (in terms of the number of different soundscapes / video) issue with this dataset? Since the duration of each video varies, the reviewer thinks it would be better to sample the same amount of duration from each source video than just randomly sampling 150 hours from the entire collected dataset.
- Does the proposed model (not the unimodal version) always require image / audio paired data for conditioning? What happens if one of the modality is absent?

---

### Official Review · Reviewer_tbtJ · 2023-11-04

**Soundness:** 2 fair
**Presentation:** 4 excellent
**Contribution:** 3 good
**Rating:** 5
**Confidence:** 3

**Summary:**

This paper proposes a new task of "audio-visual soundscape stylization" personalization. The authors propose a self-supervised learning approach to learn soundscape stylization from in-the-wild egocentric videos. The objective and subjective evaluations show the effectiveness of the proposed method.

**Strengths:**

- The paper is clearly written and nicely presented.
- The dataset, if released to the public, would be a nice contribution to the field.
- The ablation study is thorough. The limitation and analysis section is great.

**Weaknesses:**

- The proposed task is rather constrained as such reference video/audio can be tricky to find in practice. Some discussions on the potential applications and use cases would be helpful to motivate why this task is important and worth investigating.
- The authors did not compare the proposed model against the high-relevant model proposed by Chen et al. (2022). While the authors showed that the proposed method can outperform a naive "Separate and Mix" baseline, it would be helpful to compare the proposed model against Chen et al. (2022).
- The dataset only contains 223 videos, which is small in terms of diversity. It remains unclear whether the model can generalize to out-of-distribution samples.
- While the authors claim that the system can be trained on in-the-wild videos, the dataset contains a specific type of high-quality egocentric videos. It remains unclear whether the proposed method can be easily scaled up or be trained on noisy in-the-wild videos.

**Questions:**

- (Section 4.1) "... a subset of 223 videos for training ..." -> 223 is not a large number in terms of diversity. Did you check if there is any overfitting issue, say testing the model on out-of-distribution inputs?
- (Section 4.1) "2) utilize a fine-tuned CLAP audio encoder ..." -> Did you finetune the model on the CityWalk dataset? Please clarify how finetuning works here.
- (Section 4.3) "... only the speech extracted from a separation model, which excludes the acoustic factor during stylization." -> Shouldn't the speech still contain some acoustic hints? Did you apply speech enhancement to the extracted speech?

Here are some other comments and suggestions:

- (Section 1) "using the other audio-visual clip as a conditional hint." -> This is a rather constrained setup as such a reference audio/video can be tricky to obtain in practice.
- (Section 2) "For visual-based methods, ..." -> Some recent image/video-to-audio systems are missing here, e.g., SpecVQGAN (Iashin and Rahtu, 2021), Im2Wav (Sheffer and Adi, 2023), CLIPSonic (Dong et al., 2023) and V2A-Mapper (Wang et al., 2023).
- (Section 3.1 and Figure 3) The term "one-to-many" here is rather confusing as the different stylizations are the results of different input reference videos. I wouldn't call this one-to-many stylization.
- (Section 4.1) "Given the challenge of accurately modeling acoustic soundscapes in outdoor scenes using the previous methods (Chen et al., 2022; Steinmetz et al., 2022), especially in our in-the-wild setup, ..." -> I can't understand the rationale here. Chen et al. (2022) seems like the most appropriate model to compare against.
- (Figure 5) The "Separate & Mix" examples don't look right to me. They should contain the full input audio, but it's not the case for the first and last examples. If I understand it correctly the "Separate & Mix" baseline is basically "input audio + source-separated reference audio". Please clarify this.

---

### Author Response · Authors · 2023-11-17
**General Response**

We thank the reviewers for their comments and time.

## Constrained Setting Due to Limited Reference Video or Audio
Our method, similar to Du et al. [1], requires two clips from the same video only at training time. At test time, we still retain the flexibility to substitute the conditional example with a completely different audio-visual clip. This high-level idea is depicted in Figures 2 and 3 of our paper.

## Concerns on Data Diversity
Our dataset is derived from YouTube, where each video is long-form and is not confined to a specific place. It therefore provides more complex real-life scenes than the one used in Chen et al. [2]. To further demonstrate this, we will introduce a section on generalization in the revised version of our paper.

## Reconstruction Metric
Here we show mean square error (MSE) in the magnitude spectrogram using the test set with ground truth.
| Captioning (aud) | Captioning (sfx) | Captioning (img) | AudioLDM | Separate & Remix | Ours   |
|------------------|------------------|------------------|---------------|------------------|--------|
| 2.34             | 2.09             | 2.23             | 1.37          | 1.02             | **0.54** |


## Comparison with Chen et al. 2022
The model (AViTAR) proposed by Chen et al. [2] is incompatible with our setting, because they focus on generating indoor room impulse responses, while our model can also generate ambient sound. To address this gap, we retrain and compare their model under our framework. The results are presented below. We find that our method outperforms AViTAR across different metrics.
| Method | MSE* (↓) | FD (↓) | FAD (↓) | KL (↓) | IS (↑) | AVC-IB (↑) | AVC-L3 (↑) | OVL (↑) | RAM (↑) | RAC (↑) | RVI (↑) |
|--------|-----------|----------|-----------|----------|----------|--------------|--------------|-----------|-----------|-----------|-----------|
| AViTAR | 0.76      | 7.44     | 3.02      | 0.68     | 1.47     | 0.14         | 0.87         | 3.32 ± 0.11 | 3.48 ± 0.07 | 3.39 ± 0.12 | 3.40 ± 0.06 |
| Ours   | **0.54**  | **5.13** | **1.64**  | **0.59** | **2.03** | **0.17**     | **0.92**     | **3.68 ± 0.14** | **3.72 ± 0.08** | **3.55 ± 0.09** | **3.59 ± 0.06** |

## Adjust SNR in Separate & Remix
While the Separate & Remix baseline does not adjust for SNR, we note that this process can be subjective. As shown in Table, we explore the implications of SNR adjustment on corner cases such as those found in Figure 5 of our paper. We observe that SNR can help somehow but not that much. Additionally, we emphasize that our method obviates the need for such adjustments, underlining its robustness.
| Method    | MSE *(↓)* | FD *(↓)* | FAD *(↓)* | KL *(↓)* | IS *(↑)* | AVC-IB *(↑)* | AVC-L3 *(↑)* |
|-----------|-----------|----------|-----------|----------|----------|--------------|--------------|
| SNR=5     | 1.04      | 8.63     | 3.37      | 0.72     | 1.53     | 0.12         | 0.81         |
| SNR=8     | 1.02      | 8.65     | 3.34      | 0.71     | 1.51     | 0.12         | 0.83         |
| SNR=10    | 1.03      | 8.68     | 3.38      | 0.70     | 1.52     | 0.12         | 0.84         |
| Origin | 1.02      | 8.87     | 3.48      | 0.70     | 1.51     | 0.11         | 0.82         |

## Limited Technical Contribution
Our main contribution lies in introducing a self-supervised learning procedure that can restyle speech to match different audio-visual scenes, *not* a model architecture. That said, our method *significantly differs* from algorithm-centric papers like VAE, LDM, and HiFi-GAN. As the first to apply an LDM for stylizing soundscapes on in-the-wild and unlabeled audio-visual data, we believe our contribution fills a unique niche in the field.

## Training Paradigm of HiFi-GAN
Regarding HiFi-GAN's training paradigm, the intrinsic link between the input (mel spectrogram) and the output (original waveform) via short-time Fourier transform (STFT) obviates the need for external annotations. We therefore categorize this as self-supervised learning.

## Reference
[1] Yuexi Du, Ziyang Chen, Justin Salamon, Bryan Russell, and Andrew Owens. Conditional Generation of Audio from Video via Foley Analogies. In Proc. CVPR, 2023.

[2] Changan Chen, Ruohan Gao, Paul Calamia, and Kristen Grauman. Visual Acoustic Matching. In Proc. CVPR, 2022.